# Phosphorylation in the Charged Linker Modulates Interactions and Secretion of Hsp90β

**DOI:** 10.3390/cells10071701

**Published:** 2021-07-05

**Authors:** Lorenz Weidenauer, Manfredo Quadroni

**Affiliations:** Protein Analysis Facility, Faculty of Biology and Medicine, University of Lausanne, 1015 Lausanne, Switzerland; lorenz.weidenauer@unil.ch

**Keywords:** heat shock protein 90, chaperone, charged linker, phosphorylation, interactome, proteomics

## Abstract

Hsp90β is a major chaperone involved in numerous cellular processes. Hundreds of client proteins depend on Hsp90β for proper folding and/or activity. Regulation of Hsp90β is critical to coordinate its tasks and is mediated by several post-translational modifications. Here, we focus on two phosphorylation sites located in the charged linker region of human Hsp90β, Ser226 and Ser255, which have been frequently reported but whose function remains unclear. Targeted measurements by mass spectrometry indicated that intracellular Hsp90β is highly phosphorylated on both sites (>90%). The level of phosphorylation was unaffected by various stresses (e.g., heat shock, inhibition with drugs) that impact Hsp90β activity. Mutating the two serines to alanines increased the amount of proteins interacting with Hsp90β globally and increased the sensitivity to tryptic cleavage in the C-terminal domain. Further investigation revealed that phosphorylation on Ser255 and to a lesser extent on Ser226 is decreased in the conditioned medium of cultured K562 cells, and that a non-phosphorylatable double alanine mutant was secreted more efficiently than the wild type. Overall, our results show that phosphorylation events in the charged linker regulate both the interactions of Hsp90β and its secretion, through changes in the conformation of the chaperone.

## 1. Introduction

Heat shock protein 90 (Hsp90) is an important family of highly conserved chaperones responsible for the proper folding, the stabilization, and the targeting for degradation of many proteins [1,2,3,4]. It is involved in numerous cellular processes and Hsp90 regulation is critical to coordinate its activity. The increasing complexity of Hsp90’s environment during evolution required additional cellular strategies to accommodate and regulate Hsp90 chaperone function [5].

There are two Hsp90 isoforms in the human cytosol, Hsp90α and Hsp90β [6,7]. The β isoform is considered to be the constitutively expressed isoform, while the α isoform is considered to be the stress inducible one, for example upon heat shock. Regulation of Hsp90 chaperone function is mediated through protein-protein interactions and post-translational modifications (PTMs) [8,9,10,11]. A set of proteins assisting Hsp90, the co-chaperones, allow for the fine-tuning of Hsp90 chaperoning activity and its interactions with the substrate proteins, usually referred to as the clients [12]. A complex set of PTMs is known to occur on Hsp90 which, together, concur to modulate its interactions and activity, resulting in a “chaperone code” analog to the one known for histones [13].

The Hsp90 protein is composed of three main domains: an N-terminal (N) domain with an ATP binding pocket and ATPase activity, a middle (M) domain, mostly involved in binding co-chaperones and clients, and a C-terminal (C) domain which mediates dimerization and possesses a EEVD motif involved in the binding of tetratricopeptide repeat domains. The charged linker (CL), absent in bacteria but present in eukaryotes, is a very peculiar stretch of charged residues connecting the N and M domains [14]. The CL influences N versus M domain conformational rearrangements, regulates affinity for ATP, ATPase activity, protein-protein interactions, extracellular secretion and Hsp90 function in general [15,16,17,18,19,20,21,22,23,24,25,26]. Even if it has been shown to be dispensable for viability in yeast, a minimal linker length is assumed to be necessary for optimal Hsp90 function [17,19,27,28]. By comparison, the *E. coli* ortholog HtpG, which lacks the CL, has a much more compact structure with little interdomain flexibility. In HtpG, the region spanning the CL insertion folds as an antiparallel β-sheet, with the CL insertion located between β-strands 8 and 9 [29].

The structure of eukaryotic CL remains elusive since the linker is mostly not resolved on crystal structures and is thought to be highly exposed to solvent. An early study suggested that the CL in chicken Hsp90 forms two α-helices separated by a proline-containing loop [30]. A study partly supported this finding: in yeast Hsp90, two regions with residual secondary structure (RSS) separated by a proline containing loop were detected [31]. The region with RSS at the C-terminal side of the loop, which includes the corresponding β-strand 9 in HtpG, can interact with and increase the exposure of β-strand 8, which serves as a contact site for some Hsp90 clients such as p53. The CL is also able to dock onto the N domain and decrease its rotational freedom [20].

Specific features in the CL sequence are known to regulate Hsp90 function and global conformation, but the exact residues exerting these effects have not been determined [19,20,31]. The CL in yeast contains four prolines, but only one is conserved in chicken and humans, in the flexible middle loop. Conversely, three serines are present in the human sequence, while none are found in yeast. While sequence alignments are difficult due to the different length of the CL, it is clear that P218 in yeast is replaced by S226 in chicken and human Hsp90β. Two other serines are present in the CL of human Hsp90β, S255 and S261, within the region interacting with β-strand 8.

Early evidence suggested that 226 and S255 in Hsp90β and the corresponding S231 and S261 in Hsp90α are constitutively phosphorylated in humans [32]. Similarly, Hsp90 was identified as early as 1982 as a major cellular phosphoprotein in chicken liver and embryo fibroblasts [33,34]. Reports on rat and pig Hsp90 are in line with this view [35,36].

The sequences surrounding S226 and S255 present a typical casein kinase II (CK2) substrate motif and indeed CK2 can phosphorylate the CL of cytosolic Hsp90β in vitro [32]. Studies using CK2 knock-out cell lines and inhibitors have however questioned CK2 as the sole kinase responsible for these PTMs in vivo [37,38]. It has been reported that cells expressing leukemogenic tyrosine kinases (such as Bcr-Abl expressing K562 cells) have lower phosphorylation of the CL but the evidence is mixed [39,40,41].

So far, a general mechanism of regulation for CL phosphorylation has not been described. Large scale phosphoproteomics studies typically highlight S226 and S255 as quantitatively prominent phosphorylated sites (www.phosphosite.org, accessed on 17 May 2021) but do not provide a coherent regulation model. Interestingly, early studies suggested that the phosphorylation turnover of Hsp90 is slow in HeLa cells under normal conditions, but more rapid during heat shock [32,42]. One of the most recent result is the observation that the phosphorylation of S255 in Hsp90β in tissues derived from patients with cholangiocarcinoma (CCA) varies between tumor stages [43].

Phosphorylation of S226 and S255 was reported to regulate a few protein-protein interactions [39,44,45]. Unphosphorylated Hsp90 has increased binding to Apaf-1 and AhR, whereas phosphorylated Hsp90 can interact with PXR. CK2 phosphorylation on S226 and S255 in Hsp90β and on S13 in CDC37 was suggested to destabilize the Hsp90-CDC37 complex in presence of ADP [46]. Finally, in Hsp90α, mutating the homologous residues S231 and S263 to alanines reduced the activity of the client hTERT [47].

Considering the reported importance of the CL for Hsp90β chaperone activity, we speculated that S226 and S255 phosphorylation impacts more protein-protein interactions and maybe more functions than those already reported in the literature.

We thus set up a method using liquid chromatography coupled to tandem mass spectrometry (LC-MS/MS) to determine the phosphorylation occupancy (the percentage of phosphorylated residues) of S226 and S255 in Hsp90β in several human cell lines, and investigate with proteomic tools if their mutation to alanines impacts protein-protein interactions with Hsp90β. We also investigated if the mutations alter Hsp90β binding to ATP/ADP or its global conformation. Furthermore, we searched for conditions that regulate S226 and S255 phosphorylation to provide clues on possible biological function of these post-translational modifications.

## 2. Materials and Methods

### 2.1. Plasmids

Plasmids were generated using standard procedures and verified by sequencing. pCMV3-HA-Hsp90β^WT^ (“WT plasmid”) was ordered from SinoBiological Inc. (Beijing, China; HG11381-NY), which served as template to produce pCMV3-Hsp90β^S226A/S255A^ (“AA plasmid”) and no tag control plasmids. 

For pCMV3-Hsp90β^S226A/S255A^, we first generated the single mutant plasmid pCMV3-HA-Hsp90β^S226A^ by (1) using primers A1 and A2 (see Table 1) to amplify the C-terminal DNA fragment with the S226A mutation, (2) using primers A3 and A4 to amplify the N-terminal DNA fragment with the S226A mutation, (3) doing an overlap extension PCR with the fragments obtained in (1) and (2) to produce the entire Hsp90β^S226A^ DNA fragment, and finally (4) inserting the final product back into the original plasmid. We then used the primer pairs A2/A5 and A3/A6 to introduce the S255A mutation in the same way to obtain the double mutant plasmid pCMV3-HA-Hsp90β^S226A/S255A^. 

For pCMV3-Hsp90β^WT^ (no tag control), we generated a KpnI-ATG-Hsp90β^WT^ N-terminal DNA fragment with primers A7 and A4 (the point mutation is removed upon digestion of the fragment for insertion) to replace the N-terminal sequence with the HA-tag in the original plasmid.

### 2.2. Mammalian Cell Culture

HEK293T-Hsp90βKO19 cells from Didier Picard’s lab were grown in DMEM (Thermo Fischer Scientific, San Jose, CA, USA; 41966-029) supplemented with 10% (*v*/*v*) FBS (Pan Biotech, Aidenbach, Germany; P40-37500), 100 units/mL penicillin and 100 μg/mL streptomycin (Thermo Fischer Scientific 15140-122), at 37 °C under 5% (*v*/*v*) CO_2_ [48]. 

For Stable Isotope Labeling with Amino acids in Cell culture (SILAC), HEK293T-Hsp90βKO19 cells were labeled with light and heavy isotopes of lysine and arginine [49]. Base growth medium was prepared with DMEM for SILAC (Thermo Fischer Scientific; 88364) supplemented with 10% (*v*/*v*) dialyzed FBS (Merck, Darmstadt, Germany; F0392), 100 units/mL penicillin and 100 μg/mL streptomycin. Lysine and arginine isotopes were added as follows: 150 mg/L ^12^C_6_^14^N_2_-L-lysine (Merck; L5751) and 50 mg/L ^12^C_6_^14^N_4_-L-arginine (Merck; A6969) to light medium, 150 mg/L ^13^C_6_^15^N_2_-L-lysine (Cambridge Isotope Laboratories Inc., Tewksbury, MA, USA; CNLM-291-H-0.1) and 50 mg/L ^13^C_6_^15^N_4_-L-arginine (Cambridge Isotope Laboratories Inc.; CNLM-539-H-0.1) to heavy medium. 200 mg/L unlabeled proline (Merck; P5607) was added as excess in all SILAC media to limit the arginine to proline metabolic conversion. A labeling efficiency greater than 95% was confirmed by mass spectrometry.

For occupancy experiments, AsPC-1 (ATCC, Manassas, VA, USA; CRL-1682) and MDA-MB-468 (ATCC; HTB-132) cells were grown in DMEM/F12 medium (Thermo Fischer Scientific; 31331-028). OCI-Ly1 (DSMZ, Braunschweig, Germany; ACC 722) cells were grown in IMDM medium (Thermo Fischer Scientific; 12440-053). K562 cells (ECACC Public Health England, Salisbury, UK; 89121407) were grown in RPMI1640 medium (Thermo Fischer Scientific; 61870-010). HMEC (ATCC; PCS-600-010) cells were grown in HMEC medium (ATCC; PCS-600-030) supplemented with HMEC growth kit (ATCC; PCS-600-040). All media except for HMEC were supplemented with 10% (*v*/*v*) FBS, 100 units/mL penicillin and 100 μg/mL streptomycin. All cells were grown at 37 °C in a humidified incubator under 5% (*v*/*v*) CO_2_. For the heavy labeled reference cells, K562 cells were grown in RPMI160 for SILAC (Thermo Fischer Scientific; 88365) supplemented with 10% (*v*/*v*) dialyzed FBS, 100 units/mL penicillin, 100 μg/mL streptomycin, 100 mg/L ^13^C_6_^15^N_2_-L-lysine, 100 mg/L ^13^C_6_^15^N_4_-L-arginine, and 180 mg/L unlabeled proline. Cells were frozen as 30-50 μL pellets when they reached near maximal confluence or density (80% confluence for adherent cells and 0.8 million/mL density for suspension cultures) right after two washes with 15 mL cold PBS. Cells were not tested for mycoplasma contamination before experiments.

### 2.3. Cell Transfection

HEK293T-Hsp90βKO19 cells were transfected using the calcium phosphate method [50]. Plasmids were added to one volume of 250 mM CaCl_2_ (Merck; 21097), the solution was then added drop by drop to another volume of HBS 2× buffer (50 mM HEPES (Merck; H4034), 280 mM NaCl (Merck 1.06404, 1.5 mM Na_2_HPO_4_ (Merck; 1.06580)) while gently shaking. The mixture is incubated at room temperature for 1 min and is then added drop by drop to the cells. 0.1 μg plasmid DNA was used per square centimeter of cell culture. After 12 h, cells are washed once with warm PBS and put into fresh medium at 37 °C under 5% (*v*/*v*) CO_2_ for at least 24 more hours.

### 2.4. Cell Lysis of Transfected Cells for Co-Immunoprecipitation of HA Tagged Hsp90β

SILAC and non-SILAC HEK293T-Hsp90βKO19 cells transfected with pCMV3-HA-Hsp90β^WT^, pCMV3-HA-Hsp90β^S226A/S255A^, or pCMV3-no tag-Hsp90β were harvested 36 h post-transfection using trypsin-EDTA (Thermo Fischer Scientific; 25200-056), washed twice with cold PBS, and frozen in liquid nitrogen. The obtained cell pellets are lysed in 2.5 volumes “NP40 lysis buffer” (1% NP40 (Merck; 74385), 20 mM HEPES pH 7.5, 50 mM KCl (Merck; P4504), 5 mM MgCl_2_ (PanReac AppliChem, Darmstadt, Germany; A1036) with protease (Roche, Mannheim, Germany; 04693159001) and phosphatase (Roche; 04906837001) inhibitors added) during 30 min incubation on ice with periodic vortexing. Extracts are centrifuged at 16,000× *g* for 10 min at 4 °C. Supernatants are then collected, total protein content quantified by Pierce protein assay (Thermo Fisher Scientific; 22660) and expression of transfected protein measured by capillary western blot (CWB) on a JESS instrument (ProteinSimple, San Jose, CA, USA). HA tag quantification was carried out with an anti-HA antibody (BioLegend, San Diego, CA, USA; 901502) and fluorescence detection using default parameters provided by the manufacturer.

### 2.5. Analysis of Extracts of Transfected HEK293T-Hsp90βKO19 Cells (“Input” of Co-immunoprecipitations)

For input analysis, 50 μg of the same protein extracts used for co-immunoprecipitations were diluted 10× in “miST lysis buffer” (1% sodium deoxycholate (Merck; 30970), 30 mM Tris (PanReac AppliChem; A1086) pH 8.6, 10 mM DTT (PanReac AppliChem; A1101)) and digested with trypsin and LysC with a protocol adapted from Kulak et al. (“miST digestion protocol”) [51]. For SILAC samples, WT and AA were mixed in the dilution. Briefly, samples are boiled at 95 °C for 5 min, then reduction and alkylation of cysteines is done by adding 32 mM chloroacetamide (Merck; C0267) and incubating for 45 min at room temperature in the dark. Samples are then each digested during 2 h at 37 °C with a first addition of trypsin/LysC mix (Promega, Madison, WI, USA; V5073) at the beginning and a second addition of trypsin/LysC mix after one hour of digestion. After digestion, 2 volumes of 1% trifluoroacetic acid (TFA; Merck; 91699) in isopropanol (Merck; I9516) are added and the mixture is vortexed 10 s before centrifugation at 2300× *g* for 30 s. The peptides are then desalted on MCX Oasis columns (Waters, Milford, MA, USA; 186001830BA) thrice with 1% TFA in isopropanol and once with “solvent A” (2% (*v*/*v*) acetonitrile (Merck; 1.00029), 0.1% (*v*/*v*) formic acid (Merck; 1.00264)). Peptides are then eluted in three steps with first 200 μL “SCX125 buffer” (125 mM ammonium acetate (Merck; 73594), 0.5% (*v*/*v*) formic acid, 20% (*v*/*v*) acetonitrile), second with 200 μL “SCX500 buffer” (500 mM ammonium acetate, 0.5% (*v*/*v*) formic acid, 20% (*v*/*v*) acetonitrile), and third with 200 μL 0.25% NH_3_ (Merck; 5432) in 80% (*v*/*v*) acetonitrile. Peptides are dried in a speedvac and finally resuspended in “HPLC loading buffer” (2% (*v*/*v*) acetonitrile, 0.05% (*v*/*v*) TFA) before injection.

For LC-MS/MS analysis, samples were injected on an Orbitrap Fusion Tribrid mass spectrometer (Thermo Fisher Scientific) interfaced through a nano-electrospray ion source to an Ultimate 3000 RSLCnano HPLC system (Dionex, Sunnyvale, CA, USA). Peptides were separated on a reversed-phase custom packed 40 cm C18 column (75 μm ID, 100 Å, Reprosil Pur 1.9 μm particles, Dr. A. Maisch HPLC GmbH, Ammerbuch-Entringen, Germany) with a 4–76% acetonitrile gradient in 0.1% formic acid (140 min gradient except for SILAC samples separated by 95 min gradient). Full MS survey scans were performed at 120,000 resolution. A data-dependent acquisition (DDA) method, controlled by Xcalibur software (Thermo Fisher Scientific), was used to select precursors in “top speed” mode with a cycle time of 0.6 s. Masses are isolated with a window of 1.6 *m/z*, fragmentation done in HCD mode with 32% energy, and fragments analyzed in the ion trap. Peptides selected for MS/MS were excluded from further fragmentation during 60 s.

### 2.6. Co-Immunoprecipitation (Co-IP) of HA Tagged Hsp90β and Sample Preparation for Mass Spectrometry (Interactome) Analysis

For label-free experiments, 500 μg of total protein extract was mixed with 10 μg anti-HA antibody (BioLegend; 901502) and diluted to 150 μL with “NP40 lysis buffer” and incubated at 4 °C overnight on a rotator. Antibody-protein complexes are purified next morning by adding 40 μL of a 50% proteinG coupled agarose beads slurry for 2 h (Merck; 17-0618-01). Beads are washed five times with “NP40 wash buffer” (0.01% NP40, 20 mM HEPES pH 7.5, 50 mM KCl, 5 mM MgCl_2_ with protease and phosphatase inhibitors added) and immunopurified proteins are eluted with three consecutive incubations in 150 μL 0.25% NH_3_ for 15 min at 4 °C. The three elutions are then pooled, dried in a speedvac, and resuspended in “miST lysis buffer” for subsequent “miST digestion protocol” but with only one final elution from the MCX Oasis columns with 200 μL 0.25% NH_3_ in 80% (*v*/*v*) acetonitrile.

For the SILAC experiment, the procedure is the same except that the heavy samples (WT) are pooled with the light samples (AA) into heavy/light replicates after eluting immunopurified proteins off the proteinG beads. No specific mixing ratio was used. Also, SILAC samples were eluted sequentially in two steps from the MCX Oasis columns using first 200 μL “SCX125 buffer” and second with 200 μL 0.25% NH_3_ in 80% (*v*/*v*) acetonitrile. The peptides were then dried in a speedvac and resuspended in “HPLC loading buffer” before injection.

For LC-MS/MS analysis, samples were injected as described for the label free input samples.

### 2.7. Mass Spectrometry Data Analysis of HA-Hsp90β^WT^/HA-Hsp90β^S226A/S255A^/Hsp90β^WT^ Co-Immunoprecipitation Experiments and Inputs

MS data were processed by the MaxQuant software (version 1.6.3.4) incorporating the Andromeda search engine [52,53]. The UNIPROT human reference proteome database of October 2017 was used (71,803 sequences), supplemented with sequences of common contaminants. FDR filtering of both peptide and protein identifications was 1% with default MaxQuant parameters. Searches allowed for two missed cleavages and protease specificity set to trypsin (K, R) with cleavage after prolines included. Carbamidomethyl on cysteines was set as fixed modification and oxidation on methionines and acetyl at the protein N-terminal as variable modification. Initial mass precursor tolerance is 20 ppm and is then dynamically adjusted by MaxQuant after recalibration to about 5–6 ppm, and fragment mass tolerance is set at 0.5 Da. For SILAC, sample multiplicity was set to 2 with ^13^C(6)^15^N(2) as label for lysines and ^13^C(6)^15^N(4) for arginines. For label-free protein quantification, either the iBAQ values (for co-immunoprecipitation (co-IP) data) or the LFQ label-free (for input data) parameter were used [54]. For SILAC, the normalized H/L ratios were used for quantification, and these were aligned on the ratio of Hsp90β for co-IP data. Values were log_2_ transformed and normalized when needed.

The MaxQuant output file proteinGroups.txt was further analyzed with the R software, GO enrichment analysis was done with the R package clusterProfiler, and figures were produced with the ggplot2 package and Inkscape [55,56,57,58,59,60,61,62,63,64,65,66,67,68,69,70,71,72,73,74,75,76,77,78,79,80,81,82,83,84,85]. Protein groups labeled as reverse hits, only identified by site, and potential contaminants were removed. Protein groups with less than two unique peptides, four peptides, or six MS/MS counts in total were also removed. For the co-IP samples, we excluded from further analysis the proteins that were not identified by MS/MS in all six replicates in at least one condition for the label-free experiments, and in all replicates for the SILAC experiment. The co-IP data was then analyzed using log_2_ transformed and bait normalized iBAQ values or H/L ratios. Missing iBAQ values were imputed only in co-IP negative control samples, in order to carry out a Student’s *t*-test with Benjamini-Hochberg correction between the wild type or double mutant samples and the control samples. Our criteria for interactor validation were an adjusted *p*-value under 0.05 and a minimal average fold-change of 3 in log_2_ scale between the WT or AA mutant samples and the negative control samples. For GO term enrichment analysis, we selected the specific interactors with a positive or negative fold change and compared them to the complete list of proteins identified in the input.

### 2.8. Total Hsp90β Occupancy Protocol

The prepared cell pellets (see mammalian cell culture) were lysed with 4 volumes “Total lysis buffer” (TLB; 50 mM Tris-HCl pH 7.4, 500 mM NaCl (Merck; 1.06404), 0.2% (*w*/*v*) SDS (PanReac AppliChem; A3942), 10 mM sodium azide (Merck; 71289), with phosphatase and protease inhibitors added) for 1 volume pellet (volume units refer to initial pellet volume). The pellet is briefly vortexed, 0.4 volume Triton X-100 (Merck; 9002-93-1) 20% (*v*/*v*) is added, the lysate is sonicated a first time on a Fisherbrand FB120 sonicator (1 min total, 1 s on/1 s off pulses, 30% amplitude), 3.6 volumes of Tris-HCl 50 mM pH 7.4 are added and the lysate is sonicated a second time (30 s total, 1 s on/off pulses, 30% amplitude).

For Hsp90β purification, 100 μL lysate was mixed with 900 μL Tris-HCl 50 mM pH 7.4 and 4 μg F-8 antibody (Santa Cruz Biotechnology Inc., Dallas, TX, USA; sc-13119X) was added for overnight immunoprecipitation. 40 μL of a 50% slurry of protein-G coupled agarose beads is added the next day for 2 more hours. Beads are then washed 5 times with 500 μL occupancy wash buffer (OWB; 0.01% (*w*/*v*) SDS, 25 mM NaCl, 50 mM Tris-HCl pH 7.4, with protease and phosphatase inhibitors). In case of the heavy labeled K562 cells, used as dephosphorylated samples, washes were done as follows: 2 washes with 500 μL OWB, 2 washes with 200 μL CutSmart buffer (New England BioLabs, Ipswich, MA, USA; B7204S) supplemented with protease inhibitors. On-beads dephosphorylation of Hsp90 was done by adding 200 μL CutSmart buffer with 1 μL CIP phosphatase (New England BioLabs; M0290) and incubating at 37 °C for 30 min, followed by 3 washes with 500 μL OWB. All beads were eluted by adding 50 μL SDS-PAGE sample buffer 2× (0.8% SDS, 20 mM Tris-HCl pH 6.8, 10% glycerol (PanReac AppliChem; A0970), 2% β-mercaptoethanol (Merck; 63689)).

For SDS-PAGE, 25 μL of eluate of the samples of interest were mixed with 25 μL of the dephosphorylated samples and the whole run on a 10% SDS-PAGE at 130 V. Staining was done with G250 Coomassie (50% (*v*/*v*) ethanol (ReactoLab SA, Servion, Switzerland; 15058), 10% (*v*/*v*) acetic acid (Carlo Erba Reagents, Val de Reuil, France; 401391), 0.25% (*w*/*v*) coomassie blue (Merck; B0770)). Gel bands were excised from 75 kDa to 150 kDa and destained with 50 mM ammonium bicarbonate (Merck; 09830) in 30% (*v*/*v*) acetonitrile. Reduction and alkylation is done in 40 mM chloroacetamide, 10 mM TCEP (Merck; 93284), 50 mM ammonium bicarbonate at 45 °C for 1 h. Gel bands are then briefly washed with excess 50 mM ammonium bicarbonate, dehydrated with excess acetonitrile, and left to dry completely under ventilation. Digestion is done in 100 μL 50 mM ammonium bicarbonate with 0.5 μg trypsin (Promega; V5113) and 0.25 μg LysC (Promega; V1671) at 37 °C overnight. Peptides are extracted next day by incubating the gel bands with 100 μL 10% formic acid for 20 min first, and then 20 more minutes with 200 μL acetonitrile. Peptide extracts were pooled by sample, frozen in liquid nitrogen and dried under vacuum in a speedvac. The dried peptides are then resuspended in 30 μL “HPLC loading buffer”.

For LC-MS/MS analysis, 5 μL of sample was injected on the same instruments as for the co-IP experiments. The LC separation was done as described for the co-IP experiments but with a 95 min gradient. The mass spectrometer was operated in parallel reaction monitoring (PRM) mode, controlled by Xcalibur software. The list of peptides, masses, ion targets, maximum ion injection times for the monitored peptides are given in Appendix A (time windows were continuously adjusted depending on the LC column used). The Hsp90β normalization peptides were selected among the unique peptides with retention times higher than CL peptides. Two CL peptides were selected for each phosphosite: one without missed cleavage, and another with one or two missed cleavage in the vicinity of (potential) phosphosites that could inhibit trypsin. Masses are isolated with a window of 1.6 *m/z*, fragmentation done in HCD mode with 32% energy, and fragments analyzed in an orbitrap with 30,000 resolution. Full MS survey scans are taken every 1.5 s with 120,000 resolution. Peptide identification is then carried out with Mascot Server 2.6 (Matrix Science, London, UK) with the following parameters: protease specificity set to trypsin with cleavage after prolines included, maximum two missed cleavages, carbamidomethyl on cysteines as fixed modification, ^13^C(6)^15^N(2) on lysines, ^13^C(6)^15^N(4) on arginines, and phosphorylation on serines and threonines as variable modifications, 10 ppm mass tolerance on the precursor, 0.02 Da mass tolerance on fragment, threshold score at *p* < 0.05) [86]. Peptide quantification was done manually in Skyline on the extracted ion chromatograms of fragments [87,88].

Occupancy was calculated as described by Olsen et al. [89]. Briefly, the H/L ratios of the phosphopeptide (x), the unmodified peptide (y), and the whole protein (z; derived from normalization peptides) are used to compute a and b values according to the following formulas:
$$a=\frac{\left(z-y\right)}{\left(x-z\right)};\;b=(\frac{x}{y})*$$
Occupancy is then calculated in the light sample using value a and in the heavy sample using value b according to the following formulas:Occupancy_light_ = $\frac{a}{\left(a+1\right)}$; Occupancy_heavy_ = $\frac{b}{\left(b+1\right)}$

### 2.9. Nuclear and Extracellular Hsp90β Occupancy Protocol

For nuclear occupancy, 1 volume of K562 cells was lysed in three volumes of cytosolic extraction buffer (CEB; 10 mM HEPES pH 7.4, 10 mM KCl, 2 mM MgCl_2_, 0.01% (*v*/*v*) NP40, with protease and phosphatase inhibitors added). The lysate is passed through a 25 ga needle three times and vortexed for 5 min at 4 °C. After 15 more minutes incubation on ice, the lysate is centrifuged 5 min at 800× *g* at 4 °C. The supernatant is transferred to a new tube and is the cytosolic extract; the resulting pellet is washed with the same volume of CEB, passed through a 25 ga needle once, and centrifuged again, 10 min at 800× *g* at 4 °C. Supernatant is discarded, the pellet is lysed with 2 initial cell pellet volumes of TLB, and then the same lysis and subsequent Hsp90β purification, SDS-PAGE, digestion and LC-MS/MS analysis procedures are used as in the total Hsp90β occupancy protocol.

For extracellular occupancy, three 50 mL cultures of isotopically, heavy labeled K562 cells at 0.6 million/mL density were used. Conditioned mediums were centrifuged at 100× *g* for 10 min at 4 °C, and the supernatant transferred to a new tube and centrifuged again. The medium was then concentrated on 30 K Amicon 15 mL filters (Merck; UFC905008) down to 750 μL and transferred to a new tube. Filter walls were washed with 750 μL “NP40 lysis buffer” and pooled with the concentrated medium. The obtained concentrated mediums were pre-cleared with additions of 40 μL of a 50% slurry of proteinA coupled agarose beads, and 40 μL of a 50% slurry of proteinG coupled agarose beads for 20 min. Subsequent Hsp90β purification, SDS-PAGE, digestion and LC-MS/MS analysis, and data analysis procedures were done as in the total Hsp90β occupancy protocol.

### 2.10. Limted Proteolysis with Trypsin 

The samples were prepared partly as described by Tsutsumi et al. [18]. Briefly, HEK293T-Hsp90βKO19 cells were transiently transfected with plasmids expressing either HA-Hsp90^WT^, HA-Hsp90β^S226A/S255A^ or without plasmid as control as described above. Lysis and IP is done as for the interactome co-IP experiments until the beads washes. Here, beads are washed twice with 500 μL “NP40 wash buffer”, and then 3 times with 500 μL “NP40 wash buffer” supplemented with 500 mM NaCl but without protease inhibitors. During the last wash, the beads are equally split in 3 fractions. Then, 40 μL ice cold 50 mM ammonium bicarbonate with either 0, 7, or 14 ng/μL trypsin are added to the beads and incubated for 6 min on ice. The reaction is stopped by adding 15 μL SDS-PAGE sample buffer 5× and heating at 95 °C for 5 min.

CWB analysis was done on a JESS instrument as for the HA-tag expression control of transfected cells with the exception that HA-tag was quantified by an anti-HA antibody conjugated to HRP (Roche; 12013819001) diluted 1:100. 

For SDS-PAGE, 20 μL of sample was run on a 6% SDS-PAGE at 130 V, and the lanes cut in 8 gel pieces from band A to 40 kDa. Digestion of the samples was done as described previously [90].

For LC-MS/MS analysis, samples were injected on a Thermo Fischer Scientific Q-Exactive Plus mass spectrometer (Thermo Fisher Scientific) interfaced through a nano-electrospray ion source to an Ultimate 3000 RSLCnano HPLC system (Dionex). Peptides were separated on a reversed-phase custom packed 40 cm C18 column (75 μm ID, 100 Å, Reprosil Pur 1.9 μm particles, Dr. A. Maisch HPLC GmbH) with a 4–76% acetonitrile gradient in 0.1% formic acid of 65 min. Full MS survey scans were performed at 70,000 resolution. The mass spectrometer was operated in data-dependent acquisition (DDA) “Top10” mode, controlled by Xcalibur software. Masses are isolated with a window of 1.5 m/z, and fragments analyzed in an orbitrap with 17,500 resolution. Peptides selected for MS/MS were excluded from further fragmentation during 60 s. Peptide identification is then carried out with Mascot Server 2.6 (Matrix Science) with the following parameters: protease specificity set to trypsin with cleavage after prolines included, maximum six missed cleavages, carbamidomethyl on cysteines as fixed modification, oxidation on methionines and phosphorylation on serines and threonines as variable modifications, 10 ppm mass tolerance on the precursor, 0.02 Da mass tolerance on fragment, threshold score at *p* < 0.05). Peptide quantification was subsequently done manually in Skyline on the extracted ion chromatograms of selected precursors.

### 2.11. Analysis of Mutant Secretion 

HEK293T-Hsp90βKO19 cells were transiently transfected with plasmids expressing either HA-Hsp90^WT^, HA-Hsp90β^S226A/S255A^ or without plasmid as control as described above. 24 h after PBS wash, the medium was collected and the cells harvested.

Samples were prepared partly as described by Cortes et al. [91]. Briefly, 10 mL medium was spun at 100 g for 10 min at 4 °C and concentrated down to 100 μL on 30 K Amicon 5 mL filters (Merck; UFC803024). Concentrated medium was transferred to a new tube and filter walls were washed with 100 μL “NP40 lysis buffer” and pooled with concentrated medium. The concentrated medium was then denatured by adding 60 μL SDS-PAGE sample buffer 5× and heating at 95 °C for 5 min. Harvested cells were lysed as for the co-IP experiments.

Samples were analyzed by CWB on a JESS instrument with default parameters. HA-tag was quantified as for the expression controls of the transfected cells and β-actin was quantified using a β-actin antibody (Bio-Techne, Minneapolis, MN, USA; MAB8929) and chemiluminescence detection.

### 2.12. Cell Treatment and Sample Preparation for Phosphorylation Analysis (TiO_2_ Enrichment)

For heat shock treatment, K562 cells were resuspended in 2 mL medium at 37 °C or 42 °C at a density of 5 million/mL and incubated for 15 min, 1 h, and 3 h in a 37 °C humidified incubator or at 42 °C in a water bath. The cells were then immediately washed with 48 mL cold PBS and frozen in liquid nitrogen.

For Hsp90 inhibitors treatment, three 150 cm^2^ culture flasks with each 30 million K562 cells at 1 million/mL density were supplemented with 160 nM Ganetespib (Selleck Chemicals, Houston, TX, USA), 620 nM 17-DMAG (Selleck Chemicals), or 1% (*v*/*v*) DMSO as control, and a third of each culture was taken after 15 min, 1 h, and 3 h of treatment. Cells were then immediately centrifuged and washed with 50 mL cold PBS before freezing in liquid nitrogen.

For serum starvation, acidosis, and H_2_O_2_ treatment, K562 cells from a 0.5 million/mL density culture were transferred into control RPMI1640 with 10% FBS or into RPMI1640 without FBS for serum starvation, or into RPMI1640 with 10% FBS at pH 6 for acidosis, or into RPMI1640 with 10% FBS and 1 mM hydrogen peroxide (Merck; 95321) for H_2_O_2_ treatment, and incubated for 6 and 24 h (for H_2_O_2_ treatment, 6 h only).

Treated cells were lysed in “miST lysis buffer” and proteins digested with the “miST digestion protocol” described above. A quarter of the digested sample is used as is for input analysis and is resuspended in “HPLC loading buffer”. The rest of the sample is used for titanium-dioxide phosphopeptide enrichment. The peptides are desalted on a Sep-Pak tC18 μElution plate (Waters) and eluted into “TiO loading buffer” (80% (*v*/*v*) acetonitrile, 2.5% (*v*/*v*) TFA, 80 mg/mL glycolic acid (Merck; 124737)). For every 10 μg of digested peptides, 1 μL of a TiO_2_ beads (ZirChrom Separations Inc., Anoka, MN, USA) slurry at 100 mg/mL is added and incubated for 15 min on a wheel at 4 °C. The beads are then centrifuged and washed as follows, with each wash using 200 μL and followed by 5 min vortexing and short centrifugation: two washes with “TiO loading buffer”, two washes with “Wash buffer 1” (70% (*v*/*v*) acetonitrile, 0.1% (*v*/*v*) TFA, 80 mg/mL glycolic acid), two washes with “Wash buffer 2” (70% (*v*/*v*) acetonitrile, 0.1% (*v*/*v*) TFA), two washes with “Wash buffer 3” (0.1% (*v*/*v*) TFA). Enriched peptides are then recovered with two consecutive incubations for 5 min in “TiO elution buffer” (50 mM Na_2_HPO_4_, 5 mM Na_3_VO_4_ (Merck; S6508), 1 mM NaF (Merck; 201154)). Eluates are finally acidified with formic acid and 20% TFA before injection.

Inputs and enriched phosphopeptides were analyzed by LC-MS/MS with the same method as described for the co-IP experiments, except that the enriched phosphopeptides were analyzed in “top speed” mode with 1.5 s cycling time and the fragments were detected in the orbitrap with 15,000 resolution. Peptide identification was then carried out with Mascot Server 2.6 (Matrix Science) with the following parameters: protease specificity set to trypsin with cleavage after prolines included, maximum nine missed cleavages, carbamidomethyl on cysteines as fixed modification, phosphorylation on serines and threonines as variable modifications (plus acetyl at protein N-terminal and oxidation on methionines as variable modifications for input samples), 10 ppm mass tolerance on the precursor, 0.02 Da mass tolerance on fragment for enriched phosphopeptides samples and 0.5 Da mass tolerance on fragments for input samples, threshold score at *p* < 0.05). Peptide quantification was subsequently done manually in Skyline on the extracted ion chromatograms of selected precursors.

### 2.13. Mutant ATP and ADP Binding Analysis with Active Site Probes

HEK293T-Hsp90βKO19 cells were transfected as described with WT and AA plasmids. Protein extraction, sample preparation and ATP and ADP binding was done using the Pierce^TM^ Kinase Enrichment Kit (Thermo Fisher Scientific; 1862511) and the ActivX^TM^ desthiobiotin-ATP and -ADP probes (Thermo Fisher Scientific; 88311, 88313). Instead of purifying labeled proteins with beads binding to desthiobiotin, the transfected proteins were purified by immunoprecipitation with 1 μg HA antibody per 100 μg of protein extract for 1 h and 30 min. Antibody-protein complex were recovered by addition of a 50% proteinG beads slurry for 1 h and 20 min and beads were washed three times with OWB. Purified proteins were eluted by adding SDS-PAGE sample buffer 2× to the beads and boiling 5 min at 95 °C. The obtained samples were then analyzed by CWB on a JESS instrument with default parameters. HA tag quantification was carried out with an anti-HA antibody (BioLegend; 901502) and fluorescence detection and desthiobiotin was quantified with streptavidin-HRP (ProteinSimple; 042-414) and chemiluminescence detection.

### 2.14. Raw LC-MS/MS Data Availability

The mass spectrometry proteomics data for the co-IP experiments have been deposited to the ProteomeXchange Consortium via the PRIDE partner repository with the dataset identifier PXD025873 (username: reviewer_pxd025873@ebi.ac.uk, password: gtQBiP4Z) [92,93]. Identifier for data for occupancy measurements and phosphorylation comparison in stressed cells are PXD025888, doi:10.6019/PXD025888 (username: reviewer_pxd025888@ebi.ac.uk, password: zmpKajPr). For the trypsin sensitivity experiment data, identifier are PXD025878 and dpo:10.6019/PXD025878 (username: reviewer_pxd025878@ebi.ac.uk, password: blo60H4K). Login to access the data is through www.ebi.ac.uk/pride, accessed on 17 May 2021.

## 3. Results

### 3.1. High Phosphorylation Occupancy for Both S226 and S255 in Hsp90β in a Range of Cell Lines

Determining the occupancy of a post-translational modification (PTM) by liquid chromatography coupled to tandem mass spectrometry (LC-MS/MS) requires the measurement of both the modified and unmodified peptides in two samples with different occupancy. The digestion of the CL with trypsin or LysC for LC-MS/MS is incomplete due to the inhibitory presence of numerous negatively charged residues (Figure 1A,B) [94]. This results in a mixture in which we chose to measure two peptides for each S226 and S255: one without missed cleavage, and one with a single or more missed cleavage(s) in the vicinity of phosphosites that may inhibit trypsin or LysC (Figure 1B). A set of other Hsp90 peptides (Figure 1C) were used for normalization. We used Stable Isotope Labeling with Amino acids in Cell culture (SILAC) and compared Hsp90β purified from unlabeled (light) cells with Hsp90β purified from heavy labeled cells and treated with phosphatase. Light and heavy samples were mixed together prior to digestion to ensure identical digestion efficiency. We tested our set-up with synthetic peptides spanning S226 and S255, and we also controlled for the presence of peptides with phosphorylation on S261, that we determined as negligible for our purposes (Appendix A, sheet “Hsp90-S255-S261-Checks”).

As shown in Figure 1D, left pane, the chromatograms for the heavy and light normalization peptides is expected to be similar, thus with a heavy to light ratio (H/L) close to 1. For the non-phosphopeptides (Figure 1D, middle pane), we expect the heavy chromatogram to be more intense, since the heavy sample is dephosphorylated in vitro, and vice-versa for the phosphopeptide (Figure 1D, right pane). The phosphorylation occupancy is derived from these three ratios [89]. The results obtained for five human cell lines grown in standard conditions are shown in Figure 1E.

Our data suggests that about 95% of both sites are phosphorylated in all tested cell lines. This implies that at least 90% of Hsp90β monomers are phosphorylated on both S226 and S255 if phosphorylation is independent (90% phosphorylated on both, 5% only on S226, 5% only S255) and at most 95% if it is coupled (95% phosphorylated on both, 5% not phosphorylated on both). We obtained very similar values in lines of breast cancer cells (MDA-MB-468), chronic myelogenous leukemia cells (K562), pancreatic adenocarcinoma cells (AsPC-1), diffuse large B-cell lymphoma (OCI-Ly1) and untransformed, primary mammary epithelial cells (HMEC). The latter have a much slower growth rate, indicating that high occupancy on S226 and S255 is not a unique feature of rapidly proliferating cancer cells.

We sought to determine the phosphorylation state of Hsp90α as well, but digestion of Hsp90α’s CL results in an even more complex mixture (Appendix A) and revealed phosphorylation on S252 on the peptides comprising S263 (Appendix A), complicating calculations. We were able to produce data suggesting a similarly high occupancy for S231 (Appendix A), but it was not possible for S263.

In essence, our occupancy measurements indicate that phospho-occupancy at S226 and S255 on Hsp90β is constitutively high in several cultured cells under normal growth conditions.

### 3.2. Charged Linker Phosphorylation Remains High in Stressed and Heat Shocked Cells

We treated K562 cells with heat shock, Hsp90 inhibitors, or stressed the cells by altering the growth conditions and compared them to untreated cells to see if CL phosphorylation correlates with Hsp90 activity. The input lysate was also analyzed to compensate for potential Hsp90 up- or downregulation.

Of all the conditions tested (42 °C heat shock; ganetespib treatment; 17-DMAG treatment; serum-starvation; cells grown in acidic conditions; oxidative stress by H_2_O_2_ added to medium), none resulted in a significant alteration of the CL’s phosphorylation of any isoform (Appendix A).

These results indicate that the phosphorylation of S226 and S255 remains stable in a variety of conditions that have an impact on Hsp90 activity and expression levels. As Hsp90 mostly exerts its function by binding to other proteins, we next studied the impact of CL phosphorylation on the Hsp90β interactome.

### 3.3. Mutation of S226 and S255 to Alanines Increases the Amount of Hsp90β Co-Purifying Proteins

We compared the proteins interacting with “wild type” Hsp90β^WT^ (further on referred to as “WT”) and with a double Hsp90β^S226A/S255A^ alanine mutant (further on referred to as “AA”) in which both S226 and S255 are mutated to alanines, by co-immunoprecipitation (co-IP) followed by LC-MS/MS. We voluntarily limited the comparison to a double mutant only, since our data indicate that roughly 90% of Hsp90β^WT^ should be phosphorylated on both S226 and S255. The WT and AA protein have an HA tag at the N-terminal and were expressed in Hsp90β KO cells, which allowed us to eliminate interference from WT endogenous Hsp90β that would most likely form heterodimers with mutants [48]. For negative control, we used WT Hsp90 without HA tag. The experiment was carried out twice, each time with 3 replicates.

The levels of bait and input proteins were comparable between the conditions (Appendix A). The phosphorylation occupancy of S226 and S255 in the transfected WT Hsp90β protein was very close to the values normally observed for the endogenous protein (Appendix A). We identified 315 specific interactors, including well-known Hsp90β interactors CDC37, Hsp70 (HSPA1), Hsc70 (HSPA8), Hsp40 (DNAJB1), and the glucocorticoid receptor, after application of stringent filtering criteria (Appendix A). 

We did not identify any protein interacting only with WT or AA. However, comparison of AA and WT co-IPs revealed substantial quantitative differences, as many proteins co-purified in greater quantity with AA (Figure 2A, Appendix A). To confirm this unexpected result, we repeated the experiment with SILAC labeling and four replicates to maximize quantitation accuracy. We focused data analysis on the the 315 specific interactors. This allowed us to use a more accurate SILAC duplex experiment with a direct comparison of WT (heavy sample) and AA (light sample).

Input lysates were comparable, but the expression of WT in the heavy cells was slightly higher than that of AA in the light cells (Appendix A). Correspondingly, more WT was immunoprecipitated compared to AA (Appendix A). Applying stringent filtering criteria, we found 83 out of the previously defined 315 Hsp90β specific interactors that were identified and quantified in all four replicates of both WT and AA (Appendix A). Unpaired *t*-test with Benjamini-Hochberg correction highlighted 69 regulated interactors (*p*-value < 0.05).

The volcano plot for the SILAC experiment, in Figure 2A, shows that a lot of well-known Hsp90β interactors were more abundant in the AA co-IPs, and co-chaperones such as p23, DNAJC7, CDC37, AHSA1, as well as DNAJB1, HSPA1, HSPA8, and the client protein glucocorticoid receptor (NR3C1) were found among the most enriched proteins in the AA co-IPs (adjusted *p*-value < 0.05). Even though the ratios were not very large in absolute terms, they were statistically significant. For abundant proteins like Hsp90α or Hsp70/Hsc70, even such mild changes imply a large difference in the numbers of chaperone complexes impacted. Annotation enrichment analysis on the proteins with increased binding to AA highlighted the terms typically associated with the Hsp90 chaperone machine (Figure 2B, Appendix A). The same analysis on the proteins with decreased binding to AA yielded no significant term. We did not detect significant differences in changes among the subgroups of Hsp90 interactors (as defined in Figure 2A). One notable exception was the co-chaperone CACYBP, which exhibited a behavior in stark contrast to all other co-chaperones, with a higher binding affinity for the phosphorylated (WT) form of Hsp90β. Label-free and SILAC data thus both indicate that mutating S266 and S255 to alanines globally increases the amount of proteins co-purifying with Hsp90β (Appendix A). The broad but relatively mild impact of alanine mutations on the Hsp90β interactome suggests that these phosphosites alter general Hsp90β activity and conformation. We thus tested if the mutations affected ATP/ADP binding and Hsp90β sensitivity to tryptic cleavage.

### 3.4. S226 and S255 Mutations to Alanine do Not Alter ATP/ADP Binding

ATP/ADP binding was estimated for the WT Hsp90β and the AA mutant in crude cell extracts using ActivX^TM^ probes (which label ATP/ADP binding sites with desthiobiotin) and detecting desthiobiotinylation by capillary Western blot analysis (CWB). Such probes have already been used in competition assays to characterize the specificity of Hsp90 inhibitors [95]. Table 2 summarizes the average and standard deviation (s.d.) of the desthiobiotin signal normalized by HA-tag signal, as well as the *p*-value resulting from a *t*-test comparing WT and AA values. Appendix A gives the signal intensities for desthiobiotin, HA-tag, and desthiobiotin normalized by HA-tag.

Both WT and AA showed greater affinity for ATP than ADP, and both appeared to be desthiobiotinylated to a similar extent. The average value for ATP binding by the AA mutant was somewhat higher than for WT, but the difference was not statistically significant, suggesting that S226 and S255 mutation to alanines does not have a dramatic effect on ATP/ADP binding.

### 3.5. S226 and S255 Mutation to Alanine Enhances Tryptic Cleavage in the C Domain

Sensitivity to proteolytic cleavage has been used to assess global changes in Hsp90 conformation [96]. Since the CL sequence has already been reported to affect Hsp90β’s sensitivity to tryptic cleavage, we investigated whether S226 and S255 mutation to alanines also had an effect [19].

CWB analysis suggests that WT is more resistant to tryptic cleavage than AA (Figure 3A), as the full-length double mutant bands intensities decrease more than WT with increasing trypsin concentration. Two cleavage products were detected near 60 and 80 kDa with an anti-HA antibody (Appendix A), and as the HA-tag is at the N-terminal, these products were probably cleaved in the C domain. No other cleavage products were detected (analysis range was down to 12 kDa), the HA-tag being most likely cut off in these. The total HA signal of the full-length protein, the 80, and the 60 kDa products in the WT sample with 14 ng/μL trypsin is about 90% of the HA signal in the WT sample with 0 ng/μL trypsin, whereas this value drops to about 50% for AA, indicating increased tryptic cleavage near the HA-tag for AA, probably in the N domain. The full-length HA-Hsp90β bands’ intensity in the samples with 7 and 14 ng/μL trypsin amounts to 55% and 20% of the signal in the sample without trypsin for WT, respectively, and 30% and 10% for AA, respectively (Appendix A).

SDS-PAGE analysis and Coomassie staining revealed other protein bands near 75, 65, and 60 kDa in the samples with trypsin (Figure 3B). Band A indicates full length HA-Hsp90β. All bands were then trypsin digested and analyzed by LC-MS/MS.

Figure 3C gives the ratio of Hsp90β’s unique peptides MS intensity in gel band A divided by the initial intensity in band A with 0 ng/μL trypsin as a measure of full length Hsp90β cleavage. The data indicates that more full length AA was cleaved than WT: the peptides’ intensities in the samples with 7 and 14 ng/μL trypsin amount to about 35% and 10% of the intensity in the sample without trypsin for WT, respectively, and 22% and 6% for AA, which is in qualitative agreement with the CWB data.

Intensity analysis of peptides spanning Hsp90β from E42 to R679 in the cleavage products bands suggests that the main position where Hsp90β was cleaved is in the C domain (Appendix A). K607 is a reported major tryptic cleavage site in Hsp90β’s C domain, resulting in 70 kDa and 12 kDa products (Figure 3D). Figure 3E gives the MS intensity of Hsp90β unique peptides in gel band A divided by their intensity in the whole lane.

The ratios of the peptides before K607 are significantly lower than those after K607 when trypsin is added. The peptides after K607 are thus less abundant in the rest of the lane compared to peptides before K607, most likely because these are part of a cleavage product smaller than 40 kDa. Taken together, the data implies that S226 and S255 mutation to alanine facilitates tryptic cleavage in the C domain and suggests that S226 and S255 phosphorylation could have the opposite effect. Since the CL region is far from K607, the differences in trypsin sensitivity are presumably caused by a change in global conformation and not by a direct inhibition of trypsin by the phosphate groups.

In conclusion, limited proteolysis suggests that suppression of constitutive phosphorylation sites results in an Hsp90β dimer that has either a permanently more open conformation or increased structural dynamics, granting trypsin easier access to cleavage sites. Conversely, the structure of the WT, predominantly phosphorylated protein could be somewhat constrained by the modification on S226, S255.

### 3.6. Both Cytosolic and Nuclear Hsp90β Are Highly Phosphorylated on CL

We next explored if nuclear Hsp90β had a different phosphorylation occupancy at S226 and S255, but occupancy in the nuclear and cytosolic fraction are in the same range (Appendix A, sheet “NuclearOccupancy-K562Cells”). 

### 3.7. Extracellular Hsp90β Shows Lower Phosphosite Occupancy

Both cytosolic Hsp90 isoforms are secreted to the extracellular space where they carry out different functions related e.g., to extracellular matrix remodeling or wound healing [97,98,99,100,101]. We purified extracellular Hsp90β (eHsp90β) from the conditioned medium (CM) of untreated K562 cells, grown in 10% FBS, harvested at half-maximum density. We used isotopically labeled K562 cells to distinguish K562-secreted Hsp90 from bovine Hsp90 in the serum. The results are presented in Table 3 and in Appendix A sheet “ExtracellularHsp90Occupancy-K562Cells”.

Occupancy of S255 was 10% lower in the CM than in the cell. The difference was smaller for S226, between 2 to 3%. The occupancy is similarly high for both sites in intracellular Hsp90β in K562 cells (Figure 1E), but there may be a difference in eHsp90β.

### 3.8. Non-Phosphorylatable Hsp90β Is Found in Higher Amounts in Conditioned Medium

Next, we tested if S226 and S255 phosphorylation has an impact on Hsp90β secretion. We transfected cells with WT or AA HA-Hsp90β or no plasmid as negative control, and used CWB analysis to quantify HA-tagged proteins in the cells and in the CM. β-actin was used as a non-secretory protein control (Figure 4).

We found lower levels of intracellular AA compared to WT when comparing equal total protein amounts (Figure 4A). In contrast, AA was more abundant in the CM compared to WT (Figure 4B). Absolute ratios (fraction of Hsp90β in CM compared to total amount of Hsp90β in the CM and in cells) were estimated for HA-Hsp90β and β-actin and are shown in Figure 4C. The ratios for HA are higher than for β-actin, consistent with regulated Hsp90β secretion. Furthermore, the ratios are higher for AA than for WT, indicating that more AA is found in the CM than WT. It is unknown if there is a secretion capacity limit for Hsp90β, but even in the situation that the limit exists and is reached, we would expect similar amounts of WT and AA in the CM since cells were transfected at the same density.

The ratio of β-actin was used to quantify the baseline of accidental protein release in the CM, which averages at 3%. By subtracting the β-actin absolute ratio to the HA absolute ratios, we estimated the fraction of secreted Hsp90β at 1.6% ± 0.9 for WT and 4.5% ± 1.6 for AA (*p*-value = 0.06, see Appendix A), suggesting that AA is secreted 2 to 3 times more efficiently than WT.

In conclusion, within the cell there is no obvious difference in CL phosphosite occupancy for cytosolic vs. nuclear/organellar Hsp90β. On the other hand, eHsp90β appears to be somewhat less phosphorylated than the intracellular protein. The transfected AA mutant is more efficiently secreted than the WT protein, suggesting the possibility of a causal relationship between a lower CL phosphorylation and secretion.

## 4. Discussion

Lees-Miller et al. first reported on phosphorylation of Hsp90 on the charged linker, measuring 1.7 moles of phosphate per mole of Hsp90 in HeLa cells, and identified the phosphorylated residues as S231 and S263 in Hsp90α, and S226 and S255 in Hsp90β [32]. They did not detect phosphorylation on S252 in Hsp90α, nor on S261 in Hsp90β. Our own results, obtained with a method targeting the residues of interest, indicate an occupancy of S226 and S255 close to 95%, which translates into 1.9 moles phosphate per mole of Hsp90, in line with Lees-Miller et al.’s results and also with results from large scale phosphoproteomics data [102].

However, other reports showed different trends. Two studies reported reduced phosphorylation of Hsp90β in cells expressing leukemogenic tyrosine kinases [39,41]. Kurokawa et al. showed that Hsp90β was not phosphorylated on S226 and S255 when incubated in lysates of Ba/F3 cells expressing Bcr-Abl. Kim et al. reported that S226 and S255 are not modified in LS174T cells but that their phosphorylation is induced by rifampin [45]. On our side, we repeatedly measured S226, S255 phosphorylation in K562 cells, which express active Bcr-Abl, and consistently found high occupancy, contrasting Kurokawa et al.’s conclusions. Furthermore, our measurements after stress treatments suggest that phosphorylation occupancy on S226 and S255 remains high and stable in a variety of conditions that impact Hsp90 function. While our data supports the view that phosphorylation at these sites is constitutive and not regulated, the discrepancies with the aforementioned reports call for consideration of alternative interpretations and/or a more subtle model. In particular, measurement of ^32^P incorporation (as used by Kurokawa et al.) can reflect phosphorylation turnover rather than total occupancy (that we determine with our technique). Changes in phosphorylation turnover have already been reported for Hsp90 after heat stress and would not be incompatible with a high occupancy [42]. Therefore, measurements of phosphorylation turnover for the CL phosphosites could be used to shed light on these apparent conflicts.

We next addressed the effect of CL modification on Hsp90 interactions. This has been addressed in specific cases but never by a large scale, unbiased approach, as presented here [39,44,45,46]. We show that AA was able to bind almost every interactor detected, including all Hsp70 family members, co-chaperones and clients, with greater efficiency than the phosphorylated WT protein. The only interactor with more affinity for phosphorylated WT Hsp90β was the co-chaperone CACYBP, which was reported to have dephosphorylating activity on Hsp90 [103]. Our results are therefore compatible with data by two studies showing that similar AA mutants have increased binding to Apaf-1 and AhR [39,44]. The breadth and uniformity of the effect of the mutations suggests that S226 and S255 phosphorylation does not regulate specific interactions, but rather impacts other general properties of Hsp90β. 

Still, a higher amount of interactors does not necessarily imply a higher biological activity. We therefore addressed ATP binding and hydrolysis, which are key steps in the Hsp90 chaperone cycle. We showed that AA has a slightly higher binding affinity for ATP, albeit the difference was not statistically significant. Since our assay is performed in a cell lysate, a possible difference correlates with the binding of more AHSA1 (which stimulates Hsp90 ATPase activity) by the AA mutant as shown by the interactome data. Both WT and AA displayed a higher affinity for ATP than ADP, a result in agreement with one study but in conflict with others [46,104,105]. The difference may be due to the experimental conditions (in a cell lysate vs. with purified proteins) and the readout used. 

Conformational dynamics are critical for Hsp90β chaperone activity. We showed that S226 and S255 mutation to alanines facilitated tryptic cleavage in the C domain and possibly in the N domain, suggesting that lack of CL phosphorylation could shift Hsp90β global conformational equilibrium towards a more open state.

Several studies proposed that Hsp90 phosphorylation, notably on the CL, is linked to client release at the end of the chaperone cycle [42,47,106,107,108]. While our data may suggest that non phosphorylated Hsp90 has a higher affinity for clients, phosphorylated Hsp90 still binds a lot of interactors. A study even pointed out that Hsp90β in Hsp90β-AhR complexes is phosphorylated on S226 and S255, amounting to a total estimate of 1.7 mole phosphate per mole Hsp90β, thus an occupancy in the range of WT Hsp90β [44]. Moreover, the CL sequence is the least conserved domain of Hsp90, and these serines have appeared in higher order organisms only, arguing a priori against their importance for Hsp90β core chaperone functions. There are thus reasons to question the hypothesis that CL phosphosites are coupled to the progression of the chaperone cycle. 

The function of S226 and S255 phosphorylation could therefore be related to cellular or organism-related functions specific to higher eukaryotes, such as localization. Our measurements showed that nuclear-localized and cytosolic Hsp90β had virtually identical CL phosphorylation states. 

By contrast, we found a 10% lower phosphorylation occupancy for S255, and a 2–3% lower occupancy for S226 in eHsp90β. The difference may be greater in reality since eHsp90β can be “contaminated” by Hsp90β released by dead or necrotic cells. To explore a potential causal relationship between S226/S255 phosphorylation and secretion, we transfected cells with WT and AA and found clearly more AA in the CM than WT, suggesting a role for CL phosphosites in regulating Hsp90 secretion. This role could be direct or indirect, with dephosphorylation facilitating C-terminal cleavage events, that have been reported by Wang et al. as a hallmark of eHsp90 [109]. CL phosphorylation may thus be part of the chaperone code, i.e., a set of hierarchical, interconnected PTM’s, that is governing Hsp90 secretion [13].

Our results on the AA mutant closely recall a previous study by Tsutsumi et al., who identified a region flanking the CL whose mutation altered Hsp90 protein protein interactions, global conformation and secretion. Specifically, alanine mutations of the conserved hydrophobic motif in β-strand 8, ^213^IxL^215^ in human Hsp90β and ^205^IxL^207^ in yeast Hsp90, dramatically impair Hsp90 secretion and chaperone activity, and alter protein-protein interactions as well as sensitivity to tryptic cleavage in the C domain [18]. The authors showed that the mutations destabilize β-strand 8 with repercussions on hydrophobic contacts within the N domain, notably with α-helix 8. Truncation of the CL stabilized β-strand 8 in the mutants, partially restoring Hsp90 secretion and activity.

Taking these findings together with our results, we can formulate a model in which S226 and S255 phosphorylation alters the transient interactions between the C-terminal part of the CL with residual secondary structure, β-strand 9 included, and β-strand 8. Phosphate groups in the CL may form salt bridges with the numerous lysine residues in proximity and rigidify the CL, while dephosphorylation may result in increased flexibility. This mechanism could regulate contacts between β-strand 8 and the surrounding secondary structures: α-helix 8, β-strand 2 and in particular β-strand 9, as β-strands 8 and 9 are separated by the CL but do interact in the closed conformations of Hsp90, forming a β-hairpin like structure [29,31,110,111]. These contacts may then modulate the exposure of local protein binding sites, Hsp90β secretion and global conformational dynamics. In addition, high occupancy of S226 and S255 may prevent excessive destabilization of β-strand 8 by an over-flexible, non-phosphorylated CL. From an evolutionary point of view, the prolines in the yeast CL may have been replaced in higher organisms by phosphorylation sites that are used to modulate the CL’s flexibility.

## 5. Conclusions

We provided insights on the phosphorylation occupancy of S226 and S255 in various conditions and environments and on the effects of the modifications on some key molecular properties. As it often happens with Hsp90, the answers we obtained did not provide an easily intelligible picture and several unknowns remain. Disentangling the different effects of CL phosphorylation will require further investigations at different levels. To test our structural model, it will be essential to probe, for example by hydrogen/deuterium exchange, if (de-)phosphorylation of the CL impacts the exposure of the secondary structures surrounding β-strand 8. On the functional side, it will be crucial to determine the chaperoning activity of Hsp90 in the presence/absence of the modifications. Characterizing the CL phosphorylation turnover in various conditions may also provide valuable information to understand the function of S226 and S255 in physiological states such as under stress conditions. Finally, it will be important to test the relevance of CL phosphorylation on secretion of Hsp90α, that is considered the major biologically active eHsp90 species. 

## Figures and Tables

**Figure 1 cells-10-01701-f001:**
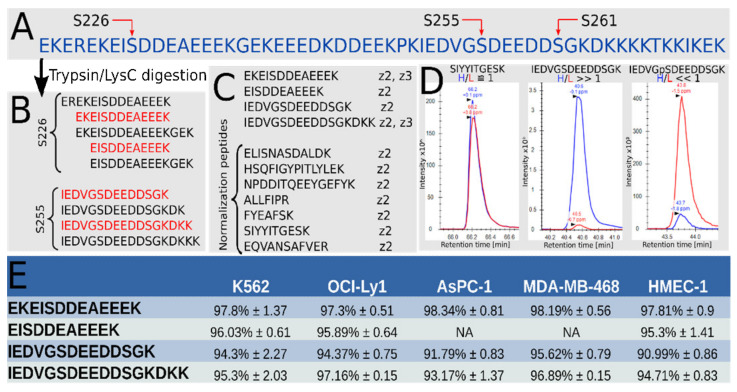
Determination of occupancy of CL phosphosites in human Hsp90β (HSP90AB1). (**A**) Amino acid sequence of Hsp90β’s CL with S226, S255 and S261 indicated by red arrows. (**B**) Typical peptides spanning S226 or S255 resulting from a tryptic digestion of the CL. The peptides marked in red were selected for targeted MS. (**C**) The complete list of peptides selected for the targeted MS method and their charge state (z number). Normalization peptides are required to determine the H/L ratio for the whole protein. Both phosphopeptides and non-phosphopeptides from the CL were measured. (**D**) Typical chromatogram for a normalization peptide (here SIYYITGESK, left pane), a non-phosphopeptide from the CL (IEDVGSDEEDDSGK, middle pane), and its corresponding phosphopeptide (IEDVGpSDEEDDSGK, right pane). The chromatogram for the heavy (H) peptide is the blue curve, and the light (L) chromatogram the red curve. (**E**) Average and standard deviation of phosphorylation occupancy for 5 human cell lines (*n* = 3).

**Figure 2 cells-10-01701-f002:**
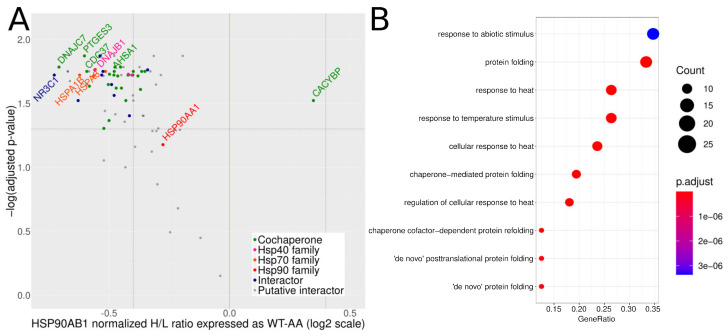
Quantitative exploration of the interactome of WT vs. AA. (**A**) Volcano plot of the inverse log_10_ of the adjusted *p*-value as a function of the normalized H/L ratio in log_2_ scale for all 83 quantified interactors (SILAC experiment). Points represent individual proteins. Point color code: co-chaperones in green, Hsp40 proteins in pink, Hsp70 in orange, Hsp90 in red, other already known interactors in blue, and putative interactors in grey. Classification of co-chaperones and known interactors is based on the “Hsp90 interactors” table by Didier Picard (www.picard.ch, accessed on 17 May 2021). Horizontal dashed line marks the 0.05 threshold for the adjusted *p*-value, vertical dashed lines mark absolute fold-changes greater than 0.4, and vertical red line marks Hsp90β. (**B**) Gene Ontology (GO) term analysis of the proteins enriched in the AA co-IP. Vertical axis: enriched GO terms. Horizontal axis: the GeneRatio is the ratio between the number of proteins enriched with the corresponding GO term and the total number of proteins identified in the input assigned to the corresponding GO term. Spot size is proportional to the count of proteins enriched (legend on the right of the plot). Spot color corresponds to the adjusted *p*-value as described in the color scale on the right of the plot.

**Figure 3 cells-10-01701-f003:**
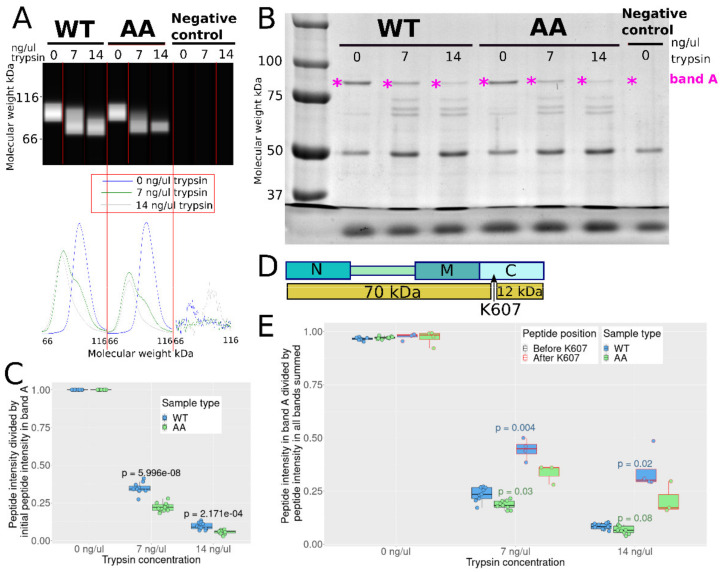
Sensitivity of WT and AA to limited proteolysis. (**A**) Reconstructed electropherogram of capillary western blot detection of HA-tag for WT and AA for increasing trypsin concentrations (top, dark background) and graph view below. (**B**) SDS-PAGE analysis of the trypsin cleavage products. Full-length Hsp90 is indicated by the pink star (“band A”). (**C**) Boxplots of the ratio of Hsp90β’s unique peptides MS intensity in gel band A relative to initial intensity in gel band A with 0 ng/μL trypsin. *p*-values comparing WT and AA samples are given above the boxes. Points represent single peptides. (**D**) The major tryptic cleavage site in Hsp90β’s C domain, near K607 and the major resulting products are shown. (**E**) Boxplots of the ratio of Hsp90β’s unique peptides MS intensity in band A relative to the summed MS intensity in the rest of the lane (whole lanes were analyzed).

**Figure 4 cells-10-01701-f004:**
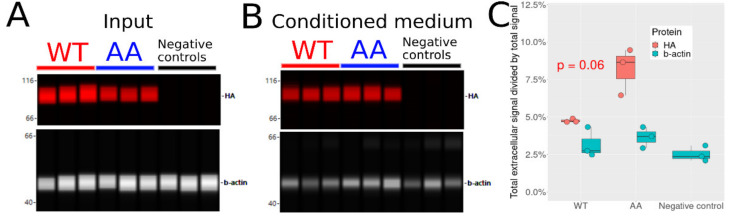
Levels of WT vs. AA in the CM of transfected HEK293T-Hsp90βKO19 cells. (**A**) CWB detection of HA-tag (red bands) and β-actin (grey bands) in transfected HEK293T-Hsp90βKO19 cells. Dual detection was performed on the same capillaries by chemiluminescence (β-actin) or fluorescence (HA-tag). (**B**) CWB analysis of HA-tag and β-actin in CM. (**C**) Total absolute CWB signal in CM divided by total absolute CWB signal (CM plus input). Total absolute signal was adjusted to account for the totality of the sample (it is the expected signal if the whole sample was analyzed at once). Points represent replicates. In red, the *p*-value from WT vs. AA *t*-test comparison of HA signal corrected by basal β-actin release.

**Table 1 cells-10-01701-t001:** List of primers used to generate Hsp90 mutants.

Primer Name	Primer Sequence (5′-3′)
A1	AAGGAAATTGCTGATGATGAGGCAG
A2 (BGH_Reverse)	TAGAAGGCACAGTCGAGG
A3 (T7_forward)	TAATACGACTCACTATAGGG
A4	CATCATCAGCAATTTCCTTCTCTCG
A5	ATGTGGGTGCAGATGAGGAGG
A6	TCATCTGCACCCACATCTTCG

**Table 2 cells-10-01701-t002:** Average and standard deviation (s.d.) of the desthiobiotin signal normalized by HA-tag signal.

Sample	Average ± s.d.	*p*-Value
WT + ATP	0.46 ± 0.03	
AA + ATP	0.53 ± 0.05	ATP:0.12
WT + ADP	0.34 ± 0.04	ADP:0.8
AA + ADP	0.35 ± 0.06	

**Table 3 cells-10-01701-t003:** Average occupancy and standard deviation (s.d.) determined for Hsp90β immunopurified from conditioned medium of K562 suspension cultures (*n* = 3). *p*-values from a *t*-test comparing the present occupancy values with K562 intracellular Hsp90β occupancy values.

Peptide	Average Occupancy ± s.d.	Adjusted *p*-Value
EKEIpSDDEAEEEK	93.3% ± 0.8	0.01
EIpSDDEAEEEK	94.5% ± 0.4	NA
IEDVGpSDEEDDSGK	85.0% ± 2.2	0.01
IEDVGpSDEEDDSGKDKK	81.3% ± 2.8	0.01

## Data Availability

The mass spectrometry proteomics data have been deposited to the ProteomeXchange Consortium via the PRIDE partner repository with the dataset identifier PXD025873, PXD025888, 10.6019/PXD025888, PXD025878 and 10.6019/PXD025878.

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
