# Peer review of "Phosphorylation in the Charged Linker Modulates Interactions and Secretion of Hsp90β"

_cells, 2021, doi:10.3390/cells10071701_

Round 1

Reviewer 1 Report

In this Weidenauer and Quadroni have identified highly phosphorylated charged linker region of human Hsp90beta, on Ser226 and Ser255. These residues were less or not phosphorylated on extracellular Hsp90 beta. Although, lack of phosphorylation did not impact the ATP/ADP binding, they did however affect Hsp90 beta interaction with the co-chaperones and clients. The limited tryptic digest of the phosphor-mutants suggest that S226 and S255 mutation to alanine facilitates tryptic cleavage in the C-domain, and phosphorylation of  S226 and S255 phosphorylation could have the opposite effect.

Overall, the paper is informative, important and highlights Hsp90 post-translational regulation or the “chaperone code”. However, the manuscript requires an extensive editing. There are few suggestion listed below that authors must take into consideration.

Title: This is too long and needs to be shorten in order to attract more readers.

Abstract- Needs to highlight the result and the importance of the study. Please avoid the use of words such as “hypothesis”. This is not a grant proposal. You need an introduction, gap in the knowledge, results and conclusion.

Result: This is OK, however, the interactome data has been mentioned briefly. This needs to be presented in as a main figure and perhaps with some bioinformatic analysis.

Discussion- This is very intense and needs to be shorten to 1-page.

Too much review of other work. The authors can relate their work to the new concept of Hsp90 chaperone code.

References: Please include these citations;

Backe SJ, Sager RA, Woodford MR, Makedon AM, Mollapour M. (2020). Post-translational modifications of Hsp90 and translating the chaperone code. J Biol Chem. 2020;295(32):11099-117.

Sager, RA, Woodford, MR, Neckers L, and Mollapour M, (2018). Detecting post-translational modifications of Hsp90. In “Molecular Chaperones: From Folding Quality to Physiological Function”, (Eds. Prince T. and Calderwood SK). Methods in Molecular Biology, Humana: London. 1709:209-19.

Reviewer 2 Report

The present manuscript by Weidenauer and Quadroni describes that a potential dynamic phosphorylation of S226 and S255 can regulate global Hsp90β conformation and impact its interaction with different partners and, consequently, its biological function. In general, the manuscript covers a relevant topic in Cell Biology with a comprehensive findings of post-translational mechanisms that regulate Hsp90 function, providing an insightful overview in this thematic. The overall topic is interesting and reaches the standards of the Cells and can be recommended for publication after minor revision.

A scheme illustrating the main findings discussed to complement and follow the narrative of the manuscript is strongly suggested.

Linguistically, there are very few typographical mistakes and grammar points to be checked and corrected in the manuscript. 

Reviewer 3 Report

In this manuscript by Weidenauer et al., the authors use several mass spectrometry-based approaches to assess the phosphorylation level of HSP90beta S226 and S255. They find that phosphorylation of these two sides is abundant in all tested cell lines and does not change in response to any tested stress. They further find that non-phosphorylatable HSP90beta S226A/S255A has a larger interactome and is secreted more into the extracellular milieu. The topic is in principle exciting and of interest to the community. The authors also do a fantastic job in disclosing experimental procedures in great details. However, significantly reducing the enthusiasm of this reviewer are missing key controls, the rather limited amount of new positive data, the lack of integration of the data into a physiologically relevant context, and the exhaustingly long introduction, results, and in particular discussion sections.  

The materials and methods section deserves special praise: the level of details is fantastic and should allow interested labs to reproduce the experiments without further input.

  • Major concerns:
  • This manuscript is at least 50% too long. It reads like a detailed PhD thesis chapter, but not like a streamlined research paper. There is a substantial need to reduce and remove unnecessary information in the intro and discussion sections. The results section should be stratified by removing detailed descriptions of how each experiment was done; this is all reflected in detail in the very nice materials and methods section. The authors should further try to use a more technical and neutral language. Elements such as “…: a technical challenge” (line 491) are unnecessary and distracting.
  • The finding that none of the treatments significantly reduced S226/S255 phosphorylation levels cultured cells raises the question if the non-modified peptide might have issues with being detected by MS, e.g. due to differences in ionization. The authors should attempt to show, using approaches other than mass spectrometry (western blotting, etc) that S226/S255 phosphorylation indeed doesn’t change upon treating the cells as described.
  • The finding that the S226A/S255A mutant pulls down interactors more efficiently is interesting. However, no efforts were made to put these exciting results into a physiologically relevant context, nor has any of the proposed different interactions been validated by e.g. western blot. The authors are also unclear if there are any interactors that only bind to either the wildtype or the AA HSP90beta protein. This is usually the first question to be addressed in such experiments. The authors should at the very least discuss these results in detail in the discussion section. They could also use GO analysis or network analysis (e.g. STRING) to get an idea how changes in interactome could alter HSP90beta-dependent processes.
  • The finding that HSP90beta binds ATP more efficiently than ADP is contradictory to published work from many labs. This should be addressed actively in the discussion.
  • Results are not discussed in relation to its potential biological relevance: e.g. are the 10% reduction in S255 and the 2-3% reduction in S226 phosphorylation in CM versus intracellular HSP90beta biologically relevant? If unknown, how could/should this be addressed?
  • The discussion section presents many interesting thoughts on HSP90beta regulation and the role of PTMs in this process. However, much of it is speculative in nature and not supported by the presented data, hence unnecessary.

Minor concerns:

  • For mass spec data searches, please disclose search parameters: Number of missed and/or non-specific cleavages permitted; List of all fixed modifications (including residue specificity) considered; List of all variable modifications (including residue specificity) considered; Mass tolerance for precursor ions; Mass tolerance for fragment ions; Threshold score/Expectation value for accepting individual spectra; Estimation of false discovery rate (FDR) and how calculated (for large datasets)
    Results section 3.1 contains mainly experimental details rather than results. Fig. 1A and 1B should be moved to the supplementary. Section 3.1 should be replaced with 3-5 sentences in the main text and the rest moved to the materials & methods section.
  • Figure 1D and corresponding text introduces a heavy to light ratio. Figure 1E indicates % of phosphorylation occupancy. The calculation how to get from heavy/light ratios to phosphorylation occupancy is not disclosed in the main text. This reviewer suggests sticking with the heavy/light ratio throughout the manuscript. Please stay coherent and use one of the two measures consequently.
  • The authors stress in the introduction that HSP90 CL phosphorylation is not understood. However, they cite 8 publications (Refs 39-47), all discussing this question. The conclusion that HSP90beta CL is poorly understood is thus hard to defend.
  • Line 560: The conclusion that: ”The occupancy seems to be independent from the type of cell line” is a bit of a stretch. The authors tested 4 cell lines and didn’t see any differences. However, there are many more cell lines & cell types out there. Please tone down.
  • Line 584: This reviewer does not understand how TiO2-mediated enrichment of phospho-peptides should help to define the ratio between unmodified/modified peptides. It may allow to compare modified peptides in sample #1 vs modified in sample #2. However, such differences could also be the result of changes in total cellular HSP90beta levels. The authors should show that TiO2-based phosphopeptide enrichment does not introduce bias by comparing this method with enrichment-free setups in a few cell lines.
  • Line 93: Does the CL domain contain more serines then S226, S255 and S261? If so, are these phosphorylated and/or linked to HSP90beta function/cellular processes?
  • Line 105: TNM is not defined
  • Lines 115-123: This sentence construct is hard to read and comprehend. Please rewrite.
  • Lines 557+: how/why the authors conclude that at least 80% of HSP90beta molecules are phosphorylated is unclear. Please clarify.
  • Lines 619+: The authors carefully distinguish between Hsp90beta and Hsp90alpha yet refer to other chaperones as HSP40 and Hsp70. Both HSP40 and HSP70 chaperone families comprise of more members than the HSP90 family. Please be more specific and use the appropriate chaperone terminology for HSP40s and HSP70s.
  • Lines 629+: The authors first find 315 proteins that specifically co-purify with HSP90beta wildtype and/or AA. In a validation experiment using SILAC, this number drops to 83. Does this mean the other 232 proteins were false-positives or do they still co-IP with WT/AA HSP90beta but at similar quantities?
  • Line 644+: Figure 2A & 2B are unnecessarily complicated; it’s hard to draw specific conclusions from this type of figures. The same data could be shown in a volcano plot, which is easier to comprehend.
  • Lines 682-688: this could be stated cleanly in 1-2 short sentences
  • Lines 689-693: are a repetition of elements found in the materials & methods section.
  • Line 694: CWB is not defined
  • Lines 709-718: This is a materials and methods paragraph.
  • Line 733: please disclose the negative controls used in Figure 3
  • Line 733: please switch figure panels 3E and 3D, as current 3E is used to justify why the K607 mutant is used in 3D.
  • Line 733: While the presented data support the authors claim, it cannot be ruled out that S226A/S255A mutations alter HSP90beta structure such that trypsin-based proteolysis is altered. Further, the S226A/S255A mutant is physiologically irrelevant. Ideally, the authors would compare phosphorylated and de-phosphorylated IP’ed HSP90beta in this experiment. This would allow to directly discern the effect of phosphorylation on HSP90beta tryptic processing. Alternatively, the authors could show using e.g. isothermal calorimetry or spectroscopy that the S226A/S255A mutant still folds correctly and retains a wildtype-like structure.
  • Lines 744+: Figure 3 shows the most pronounced differences for experiments using 7ng/ul trypsin. However, the accompanying results section (lines 748+) focuses on the 14ng/ul condition while not discussing the 7ng/ul condition.
  • Lines 806-808: the presented conclusion seems to be a bit of a stretch. Please tone down.
  • Line 829: Figure 4A indicates that HSP90beta wild-type is expressed more than the AA mutant. In Figure, 4B, the difference between wild-type and AA mutant levels in CM is more comparable. An alternative explanation, contrasting the one put forward by the authors, would be that the cellular machinery supporting HSP90beta secretion has a capacity limit. If so, the total extracellular HSP90beta level would be independent from medium (AA) and high (WT) HSP90beta levels
  • Line 829: This reviewer is confused by the quantitative data shown in Fig 4C: For WT, there is substantially more signal in the input as compared to the CM condition. Fig. 4C indicates that ~ 5% of HSP90beta is found extracellular, which seems low but possible. However, for AA, signal shown in Fig. 4C look rather comparable for AA input and AA CM. Thus, one would expect to see ~ 50% of HSP90beta to be in the CM. Yet, Fig. 4C claims it’s only ~7.5%. Please address.
  • Lines 913-915: the authors write: Once more, the regulation of S226 and S255 phosphorylation and its impact on interactions has only been addressed a few times in the literature, and never by a large scale, unbiased approach [41,42,44,50]⁠. This reviewer would argue that having 4 papers dealing with the phosphorylation of two specific HSP90beta residues is already a lot; many phosphosites have not been addressed at al. Please reconsider the statement.
  • Lines 918+: the authors write: The results show that, strikingly, the double alanine mutant was able to bind almost every interactor detected, including all Hsp70 family members, cochaperones and clients, with greater efficiency than the phosphorylated, wild-type protein. While the author’s conclusion is supported by their MS data, no direct client/co-chaperone binding assays have been performed to confirm this finding. The authors should perform client binding reactions using AA and WT HSP90beta such as luciferase (re-)folding assays. This would allow to read out both client binding and refolding differences between WT and AA HSP90beta in presence and absence of co-chaperones and/or HSP90beta inhibitors.
  • Lines 926-933: proposing explanations only to refute them in the same paragraph is not really necessary.
  • Lines: 990-995: This reviewer thinks that the presented data does not provide evidence for a role of S226 and S255 phosphorylation in the chaperone cycle. Such over-interpretations, or “speculative model”, as the authors put it, are unnecessary.

Round 2

Reviewer 3 Report

The authors made an honest attempt at improving the presented work, while deciding not to invest in performing additional experiments as suggested (e.g. western blots, HSP90 activity assays, etc.). Given the current COVID-19-related complications in our business, we believe it would be unfair to insist on additional work to be performed. Consequently, we support acceptance of this article.